# Evaluating a Spoken Dialogue System for Recording Systems of Nursing Care

**DOI:** 10.3390/s19173736

**Published:** 2019-08-29

**Authors:** Tittaya Mairittha, Nattaya Mairittha, Sozo Inoue

**Affiliations:** Graduate School of Engineering, Kyushu Institute of Technology, 1-1 Sensui-cho, Tobata-ku, Kitakyushu-shi, Fukuoka 804-8550, Japan

**Keywords:** dialogue system, data collection, electronic health record

## Abstract

Integrating speech recondition technology into an electronic health record (EHR) has been studied in recent years. However, the full adoption of the system still faces challenges such as handling speech errors, transforming raw data into an understandable format and controlling the transition from one field to the next field with speech commands. To reduce errors, cost, and documentation time, we propose a dialogue system care record (DSCR) based on a smartphone for nursing documentation. We describe the effects of DSCR on (1) documentation speed, (2) document accuracy and (3) user satisfaction. We tested the application with 12 participants to examine the usability and feasibility of DSCR. The evaluation shows that DSCR can collect data efficiently by achieving 96% of documentation accuracy. Average documentation speed was increased by 15% (P = 0.012) compared to traditional electronic forms (e-forms). The participants’ average satisfaction rating was 4.8 using DSCR compared to 3.6 using e-forms on a scale of 1–5 (P = 0.032).

## 1. Introduction

With an aging population, the need for nurses in nursing care centers is likely to increase. For the safety and well-being of patients, improving the efficiency and productivity of nurses is essential [1,2]. One highly cost-effective solution is using useful tools to reduce their workload. An electronic health record (EHR) can be accessed on-demand and potentially reduce medical errors, documentation time, and cost compared to the usual paper form [3,4]. A different approach that has been studied and integrated into EHR technology. For example, entering data into electronic forms (e-forms) on a smartphone application [5,6] or using speech recognition technology, which is a technology that allows spoken input into systems [7,8,9]. Speech recognition enables hands-free control of various devices and helps to capture speech at a faster rate than type speed. However, this technology is far from user-friendly and requires preprocessing technique in speech recognition, such as modifying record errors in conversion (from speech to text) or document summarization from the free text format. These processes may increase documentation time and workloads instead.

For that proposed, we present a dialogue system care record (DSCR) based on a smartphone that can accumulate and produce care records with the exact format at a fast pace using speech. DSCR allows users to record the data by talking with the system agent in natural conversation. The system detects users’ intention, recovers errors and parses into discrete data fields in the valid format. We conduct experiments in a simulated situation to examine the usability and feasibility of DSCR for recording systems of nursing care. We compare documentation speed of DSCR with e-forms of a smartphone application. The data used in experiments based on real nursing care activities, and it was preliminarily evaluated with general users with a deep intuitive understanding of the system before testing it with the nurses.

This paper is organized as follows: first, we examine the related works involving the different strategies that applied to EHR technology. Next, we introduce our proposed method behind DSCR. Then, we show the end-to-end system and describe each component in detail. After, we present a preliminary performance evaluation. Finally, we discuss the result obtained and the future direction.

## 2. Background

We first review the literature on an e-form based on smartphone use and traditional speech recognition integrating with this form, then we show how the dialogue system relates to the latter.

The future of health will likely be driven by digital transformation; the transition from paper to EHR has undoubtedly increased the efficiency and productivity of nurses [1]. EHR helps to reduce errors associated with poor handwriting and costs through decreased paperwork, enable quick access to patient records for more efficient care, share electronic information with patients and other providers. The use of smartphones is so usual nowadays that leading to rapid growth in the development of EHR software applications for these platforms [5,6]. The main advantage of using e-forms for data entry is straightforward. Data collected can be checked for accuracy at the time that it is entered by validating the required fields before acceptance. They are also useful in a general sense as graphical user interfaces such as select form, checkbox, or radio box. However, the typing speed is slow and error rate is relatively high for taking long notes.

The speech recognition technology can translate speech to text, which is commonly faster than typing, and it has a capability that allows nurses to record while they are taking care of patients (hands-free mode) or when they have jobs that occupy their hands, they would greatly benefit from a voice-controlled environment [7,8,9]. Although EHR using speech recognition success has been reported, full adoption of the system into the real care nursing practice is still in the distant future. Existing EHR applications that using this technology are not sufficient to use for all of the documentation as it applies for only note recording (free-text forms). Because it is quite complicated when the transition from one field to the next with speech commands. Moreover, weak speech recognition systems and unrecognized word utterances can hugely downgrade the performance of these systems. Therefore, these systems must handle speech recognition errors before using or feeding it to the pipeline for supervised-learning tasks that utilize pre-trained models.

The dialogue system is a software system, which can interact with a human user in natural language [10]. It is based on many component technologies such as speech recognition, natural language understanding and natural language generation. The main differences between simple speech recognition and dialogue systems is their conversational nature of interaction with users. It provides the most efficient and helpful mode of communication, which enables productive and flexible communication [11,12,13,14]. The dialogue system that can converse with a human by using voice-based recognition is called a spoken dialogue system (SDS). There is a growing interest in a conversational user interface, as they can truly enable people to be mobile and hands-free such as Microsoft Cortana [15], Amazon Alexa [16], Google Assistant [17], Google Home [18]. These devices become popular when designing deep learning-based dialogue systems, which are an attempt to converse with users in natural language. Interactive interfaces that permit a user to ask a natural language question and receive an answer, possibly in the context of a more extended multi-turn dialog. They are also much more practical for people who are multitasking or since screen fatigue is a concern, for example, nurses who need to record patient care when taking care of patients.

Here are the ways DSCR help to solve these issues. The dialogue system uses speech data as input and processes speech to text, then detects intention and extracts meaning from this input. The agent stores a few sentences describing who it is and a dialog history. When a new utterance is received from a user, the agent will combine the content of this knowledge base with the newly acquired utterance to generate a reply. With these processes, the system can handle errors and clarify ambiguous utterance before filling in forms, so that we can ensure data will be collected with the correct values.

## 3. Methods

In this section, we introduce the basic concepts of the dialogue system along with a way of collecting care records. Then, we show the activity diagram of the user request process of DSCR and describe its algorithm in detail.

### 3.1. Dialogue System

Dialogue systems, conversational agents or chatbots are software programs that support conversational interaction between humans and machines in natural language [10]. It can be based on text-based or speech-based and can also be used on different devices. Typically, dialogue systems can classify into two categories—a task-oriented dialogue system which is used in this paper; and a non-task-oriented dialogue system or chatbot. The task-oriented dialogue system is designed for a particular task and set up to have short conversations [19,20] such as booking flight tickets, talking to customer care service and asking about the weather. While a non-task-oriented dialogue system or chatbot is designed for unstructured conversational as a conversation between human and human [21,22]. The dialogue system requires an understanding of natural language in order to process user queries.

### 3.2. Proposed Method

The initial step after getting a user request in a dialogue system is to understand the intent correctly. If the dialogue system fails to understand the meaning of the user’s request, it may lead to giving an inappropriate response or no response. To understand the user’s intent correctly, we have focused on a frame-based system. The frame-based system is designed for a task-oriented dialogue system to get information from the user and complete the task more efficiently. Here the problem is similar to form filling, which asks the user questions to fill the slots (i.e., entities) in a frame (i.e., intent) and repeats until all the questions have been asked. The goal of the frame-based system is to extract two things from the user’s utterance. The first is the intent—what goals are users trying to achieve. The second is an entity—the particular entities that the user intends the system to understand from their utterance with respect to their intent [23]. The intents and entities are mapped to the terms of input e-forms. For example, measuring vital signs has a body temperature input field that requires an integer value greater than 0. An utterance can be expressed as “37 C of body temperature”, and the utterance should give an intent like (Intent = vital signs record, Record = body temperature, Value = 37 C).

Figure 1 shows an architecture for a dialogue state of a dialogue system including 4 components. Given an utterance, the natural language understanding (NLU) maps it into semantic entities. The dialogue state tracker (DST) maintains the current state of the dialogue and the entire set of entity constraints the user has expressed so far. The dialogue state represents all information that is relevant to the system and handles unexpected input from the NLU. Based on the state of the dialogue, the dialogue policy selects what the system should say next. Then, natural language generates (NLG) generates the text of the response to its surface.

### 3.3. User Request Process

The process of DSCR is slightly different from traditional frame-based dialogue systems because users are unnecessary to complete all records at the same time. Also, with this classic dialogue state, users cannot edit input instantly when they had an error input. They have to wait until finish all processes or all required questions have been asked. For that problem, we subdivide the dialogue of each record independently. In this way, we can guarantee that each record is collected in the right format. Figure 2 shows the user request process. Firstly, the user is made to enter the activity type, then the system will find the match activity type. If not found, the system will request users to enter again. User requests are redirected to the different records. The system will check the input type of each record and suggest possible values to the user if it is the incorrect format. Finally, the system will return the confirmation response to the user.

### 3.4. Data Collection

An algorithm behind DSCR for making decisions is that matching to a user utterance based on supervised machine learning models so that data resource is crucial in the development of effective intent classification model and modeling efforts in conversational. To collect relevant information and generate alternative replies when talking with users. We use a Machine-to-Machine (M2M) [24,25] paradigm to collect training data. M2M is one of the most popular data collection approaches among intelligent virtual assistants on the market nowadays. The idea is to define a set of prompts for each intent and generate dialogue templates with each prompt, then paraphrasing to natural language by a human. Figure 3 shows the M2M data collection paradigm. Following this process, we can collect all possible in dialogue flows without ambiguous semantics. In our experiments, we ask 3 users for paraphrasing jobs and ask them to write several constructive paraphrases of each prompt and keep the number of training samples per intent balanced across intents.

## 4. Implementation

We develop DSCR that runs on Google Assistant [17] to examine the usability and feasibility of the system. Figure 4 shows a high-level architecture of DSCR including the dialogue application and the dialogue web server. Each architectural component is discussed in greater detail below.

### 4.1. Dialogue System Application

The dialogue system application is the communication channel between the bot and the user. We use Dialogflow [26] for building conversational interfaces. Dialogflow is a cloud-based NLU platform that provides a web interface to create bots which makes it easy to create initial bots. It also facilitates integration with Google Assistant that provides the functionalities of automatic speech recognition for converting speech to text on-device. We used Dialogflow to handle intent classification and entity classification. Dialogflow classifies the intent by applying machine learning models. We can train a classifier by adding training phrases to map from sentences to intents and a sequence model to map from sentences to slot fillers. The training phrase is a list of possible user inputs that we expect users would say to our bot to trigger the intent). Dialogflow invokes the specific intent if the training phrases and the input contexts are matched. We created a list of intents and for each intent provide a set of training phrases representing what a typical user may say for that intent by using The Machine-to-machine (M2M) paradigm (See in Section 3.4).

### 4.2. Dialogue System Web Server

The dialogue web server is the main class that handles communication between the bot and the user. We use NodeJS to construct the back-end web server. It manages the state of the conversation and generates responses via an HTTPs callback webhook (a webhook is an HTTP callback that automatically occurs when certain things happen). All successful records were stored as JSON and synced real-time data to cloud storage. Finally, users can see all records on the web visualization dashboard.

## 5. Experimental Setup

To assess our hypotheses, we designed a series of user studies tailored to the scenarios of use, with 5 activities based on real practical nursing care. Each activity included various record types, which are explained in detail in Table 1. Each activity was tested with 12 participants—3 males and 9 females with ages ranging from 24 to 35. The participants were asked to perform 2 tasks, including (1) entering data on e-forms of a mobile application; (2) entering data by talking with the dialogue system agent on the Google Assistant application. After completing the task, participants were asked to answer a questionnaire related to their experience, satisfaction and further details. Figure 5 shows an example of screens of each application that participants were asked to complete.

## 6. Preliminary Evaluation

Here, we examine the feasibility of DSCR by evaluating documentation speed and task error rates from a set of dialogues produced by the interaction of the system with each user. Then, we explore users feedback regarding the usability of DSCR for nursing care documentation.

### 6.1. Measures of the Dialogue System Feasibility

We conducted a paired sample t-test to compare documentation speed and user satisfaction between DSCR and e-forms, all significant, with p < 0.05, two-tailed tests, with n = 12. Table 2 shows the average documentation speed in DSCR and e-forms. Participants documented the average of 27.29 (SD 2.47) second per activity when used DSCR and 32.76 (SD 2.49) second per activity when used e-forms. The overall increase in documentation speed through DSCR was 15% (P = 0.012). Figure 6 shows documentation speed in the DSCR and e-forms of each activity performed by users.

We then evaluate the quality of the DSCR by measuring error rates (see in Table 3). DSCR does not recognize user input on the average 28.2% of all utterances. When looking at errors into the conversation and the subsequent intentions invoked. We found that 24.5% of mistakes occur from speech recognition, and that lead to misinterpretation such as speakers pronunciation errors. We accept that it is usual for the development that these technologies get things wrong on occasion. However, speech recognition improvements will ensure you have satisfied users. Only 3.7% of errors arise from understanding the users’ intention. These mismatches occur when the system fails to match the user utterance to an appropriate intent, which usually is due to not matching intent training phrases or entity values. However, it is not much of a concern for our records because our methods provide the functionalities that users can validate and edit their data to make accurate before storing in the database.

### 6.2. Measures of the Dialogue System Usability

We used a 5-point Likert scale on online survey forms to ask for the users’ experience in the use of DSCR. The survey helps us to answer specific questions, and results are easy to interpret since users are arranged on a scale we have configured. Each of these questions would then have a set number of responses for users to choose from, respectively: (1 = Highly Dissatisfied, 2 = Dissatisfied, 3 = Neutral, 4 = Satisfied, 5 = Highly Satisfied). The survey contained questions about user satisfaction from using the system as follows:–Q1: Was the system easy to understand?–Q2: Did the system understand what you said?–Q3: Was it easy to record the information you wanted?–Q4: Was the pace of interaction with the system appropriate?–Q5: Did you know what you could say at each point in the dialogue?

Figure 7 shows user satisfaction relative to the usability of DSCR. We found that users were satisfied with Q1, Q3, and Q4. These indicated that the DSCR is simple enough to understand, the process of recording was relatively straightforward and the interaction appropriate. However, in Q2 and Q5, the number is smaller than we expect. We assumde that some users were dissatisfied in Q5 because we did not show the participants the task description while they were working on the task, so the main problem that users would face was not knowing what they should say to the system. While in Q2, speech recognition quality is of evident influence on user satisfaction, which shows sufficiently high error rates. We also asked participants to compare DSCR and e-forms. Participants self-assessed their satisfaction on a scale of 1–5, the same as with the other questions. Statistical analysis was done using the paired sample t-test. The participants’ average satisfaction rating was 4.8 (SD 0.39) using DSCR and 3.6 (SD 0.48), when using e-forms. DSCR use was significantly higher than the use of e-forms (P = 0.032). The main reasons are ease of setup and use.

## 7. Limitations and Future Research

### 7.1. User Interface

The main limitations of the system are the recording interface. Since users have to take the initiative record, although with this method they can reduce the activity state space since user actions are only allowed at specific points in the dialogue; however, users cannot see record details and remember what they have to record. Also, users cannot see their records in real time as they can do in the e-forms. They have to log on the smartphone and see history on the web site. Free form text fields cannot be validated immediately, and users cannot select specific position the same as they can move the caret position to delete in input forms as they have to delete all records and start recording again from the beginning. We believe combining e-forms and the dialogue system in the same system will be more helpful for improving the user experience. For example, users see the input fields of e-forms on the smart app and record with their voice. Then, the system converts speech to text and adds value to text input fields instantly.

### 7.2. Internet Connection

The current system requires an internet connection because we built it on top of Google Assistant and used Dialogflow APIs for handling conversational flows. In future works, we have planned to implement a dialogue system in our Android application (See in Figure 5). We also proposed solutions to take network connection problems into account. For example, we can use the speech recognition library (e.g., Cloud Speech-to-Text API Client Library for Java) to use offline speech recognition. We can build our classification models using on-device machine learning to classify intents and entities. We can also store data in local storage and upload data or sync to the server automatically when the phone has an internet connection.

## 8. Conclusions

In this paper, we present DSCR, the spoken dialogue system to record nursing care data. DSCR integrates speech recognition and natural language understanding through mobile technology, which helps to improve the accuracy of nursing care records compared to the system uses speech recognition without dialogue. Our evaluation data indicated DSCR increases the speed of documentation when compared to e-forms. Also, users are satisfied with DSCR as shown by the ratings given along with the review. These encourage us to develop DSCR further.

## Figures and Tables

**Figure 1 sensors-19-03736-f001:**
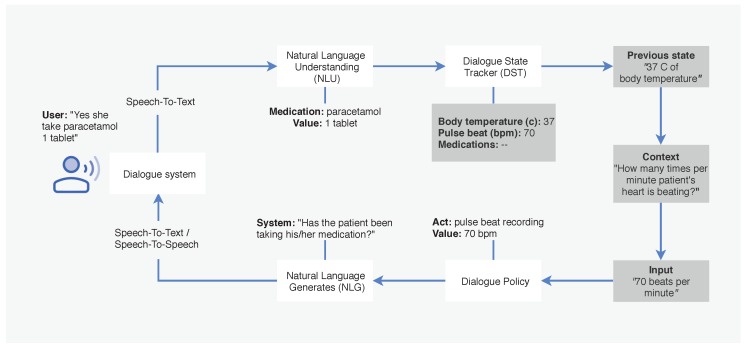
A frame-based architecture and dialogue state in a dialogue system.

**Figure 2 sensors-19-03736-f002:**
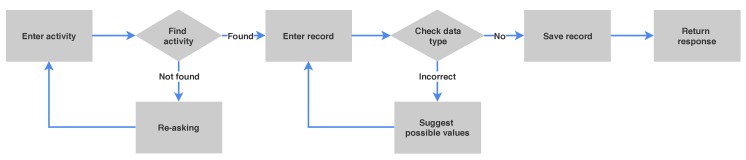
An activity diagram of user request processes of each record.

**Figure 3 sensors-19-03736-f003:**
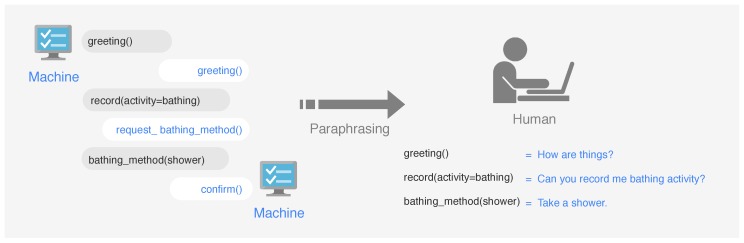
The Machine-to-machine (M2M) paradigm.

**Figure 4 sensors-19-03736-f004:**
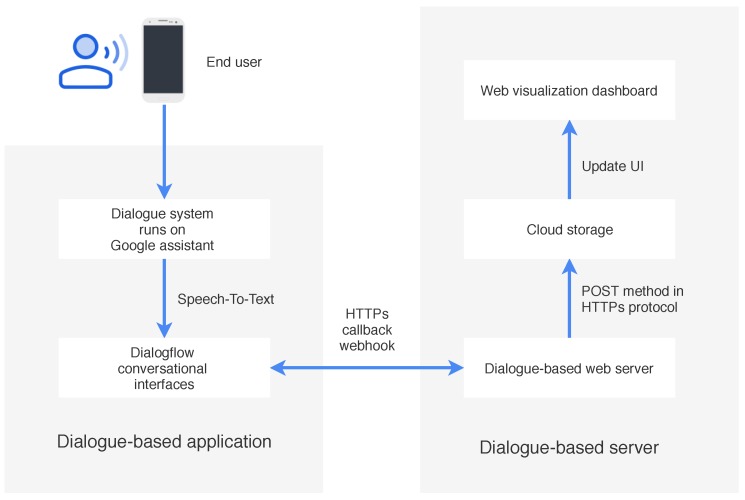
A high-level architecture of dialogue system care record (DSCR).

**Figure 5 sensors-19-03736-f005:**
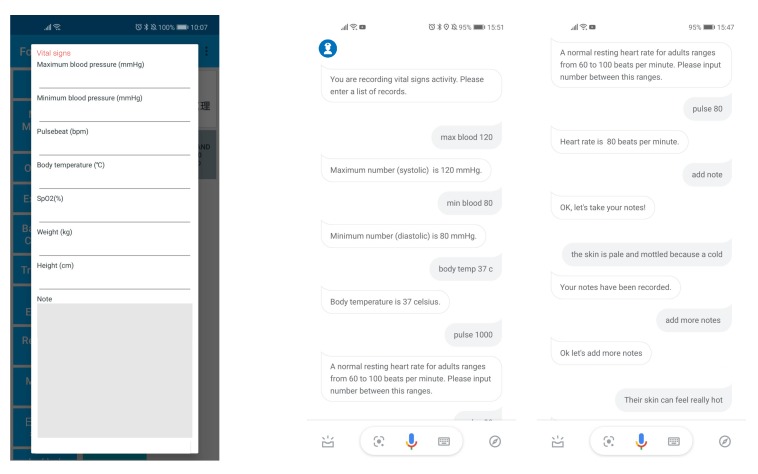
Screenshots of the e-forms on Android app (**left**) and DSCR runs on Google Assistant (**right**).

**Figure 6 sensors-19-03736-f006:**
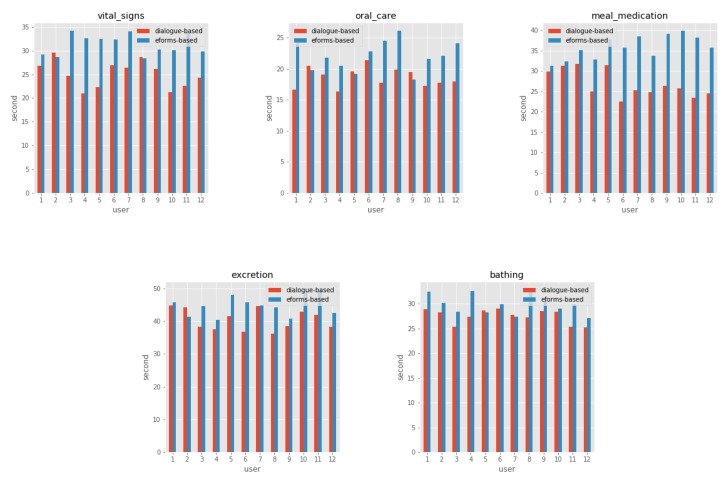
Speed in the documentation of dialogue-based and e-forms of each activity that performed by users.

**Figure 7 sensors-19-03736-f007:**
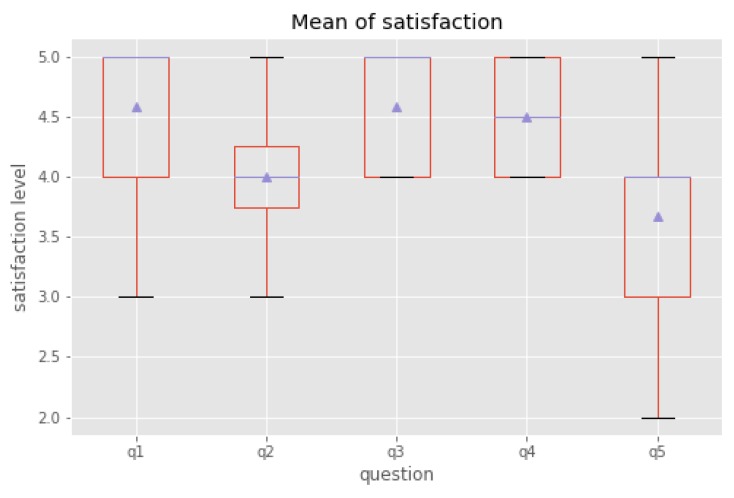
User satisfaction in the use of dialogue system. The mean is represented by the triangle and the median is the horizontal line.

**Table 1 sensors-19-03736-t001:** List of records.

Activity	Record	Type	Possible Values
Measuring vital signs	Maximum blood pressure	Input	greater than or equal to 0
	Minimum blood pressure	Input	greater than or equal to 0
	Pulse beat (bpm)	Input	greater than or equal to 0
	Body temperature (c)	Input	greater than or equal to 0
	Weight (kg)	Input	greater than or equal to 0
	Height (cm)	Input	greater than or equal to 0
	Note	Text	e.g., being sick, skin colors, have a fever, pain complaint
Meal and medication	Meal assistance	Select	self-reliance, setting only, partial care, full care
	Dietary volume	Select	0 to 10
	Meal size	Select	0 to 10
	Amount of water	Select	0 to 500
	Medication	Select	self-reliance, assistance, no medication
	Note	Text	e.g., dysphagia, appetite loss, using ThickenUp clear
Oral care	Oral cleaning	Select	self-reliance, setting only, partial care, full care, no cleaning
	Denture cleaning	Select	use of detergent, wash in water, no cleaning
	Note	Text	e.g., using sponge brush, using interdental brush, using dental floss, oral wound
Excretion	Method of excretion	Select	toilet, portable toilet, urinal, on the bed
	Excretion assistance	Select	self-reliance, setting only, partial care, full care
	Mode of Excretion	Select	defecation, urination, fecal incontinence, urinary incontinence, no excretion
	Urine volume	Select	small, medium, large, no choice
	Defecation volume	Select	small, medium, large, no choice
	Type of Waste	Select	watery mail, muddy stool, ordinary, hard stool, colo flight, no choice
	Diapering	Select	putt exchange, rehapan replacement, diaper change, wipe, vulva cleaning, change assistance
	Note	Text	e.g., hematuria, bloody stools, a tight stomach
Bathing	Bathing method	Select	general bath, shower bath, machine bath, wipe, it was planned to bathe but there was no conduct
	Bathing assistance	Select	self-reliance, setting only, partial care, full care
	Use of bath aids	Text	e.g., shower carry use

**Table 2 sensors-19-03736-t002:** Average record speed of the dialogue system and e-forms that users performed of each activity.

Experiment	Descriptive	Vital Signs	Meal	Med	Oral Care	Excretion	Bathing
Dialogue	Mean	25.07	26.8	18.61	40.45	27.53	27.69
	Std	2.84	3.32	1.57	3.23	1.42	2.47
	Min	20.97	22.47	16.3	36.12	25.2	24.21
	Max	29.54	31.7	21.32	44.76	29	31.26
E-forms	Mean	31.31	35.82	22.03	44.77	29.88	32.76
	Std	2.15	2.86	2.38	3.13	1.96	2.49
	Min	28.35	31.29	18.27	40.34	27.18	29.08
	Max	34.17	39.83	26.09	49.59	32.82	36.5

**Table 3 sensors-19-03736-t003:** Error rates of each activity in recognizing speech and in predicting intention of users.

Activity	Speech Error	Intent Error
Measuring vital signs	6.4%	0.8%
Meal & Medication	3.2%	0.8%
Oral care	4.0%	0.7%
Excretion	7.3%	1.0%
Bathing	3.6%	0.4%
Sum	24.5%	3.69%

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
