# Peer review of "Evaluating a Spoken Dialogue System for Recording Systems of Nursing Care"

_sensors, 2019, doi:10.3390/s19173736_

Round 1

Reviewer 1 Report

This study proposes a dialogue system to capture information during the point of care to collect data efficiently.

Major concerns:

1.     This is an interesting study that proposes to use chatbot to capture information for EHR system. One major concern is that due to the unsatisfactory performance of speech recognition and information extraction, is it really worthy to sacrifice the information accuracy for EHR systems?

2.     What speech recognition techniques were used?

3.     What techniques were used to extract values from the answers? Is natural language processing technique being used?

4.     Table 2. P-values should be reported to show whether there is statistical significance.

5.     Section 4. What is there is no or poor internet connection? Is there a buffering component to allow postponed uploading?

6.     Why Google Assistant and Dialogflow were chosen as the platform?

7.     The manuscript needs thorough grammar checking/proofreading. For example,

7.1  Abstract. Line 3. “is are”-->”is of”

7.2  Abstract. Line 9. “initial”-->”initialize”

7.3  Page 1. Line 27. “These process”

7.4  Page 1. Line 29. “with exact format and speedy”-->”speed”

7.5  Page 4. Line 123. “In Figure 2 shows”-->”Figure 2 shows”

Reviewer 2 Report

Summary:

In this paper, a dialogue system care record (DSCR) has been developed on a smartphone platform. They tried to examine the utility and feasibility of the system, while the evaluation shows 96% of accuracy in predicting users’ intentions and at least 15% increases in the speed of documentation compared to a simple system using electronic forms (e-forms) on a smartphone application.

Strengths:

-       Literature review has a good flow by starting from previous studies on e-form based on smartphone use leading to dialogue systems and their relations with speech recognition.  

-       They used Google assistant to examine the feasibility of the system to collect records and compared a dialogue system and e-forms applications, which is a valid sort of experiment in this domain.

-       In method section, diagrams and figures were well employed to clarify the problem and solution.

Issues:

-       There are typos and grammatical errors that need to be fixed, e.g. the 2nd and 3rd sentences in Abstract are full of errors, also section 4 and 6 have grammar issues. I recommend authors to carefully proof read all sections.

-       The number of participants in experiments is limited to 12 people which avoids the results to be generalised, in terms of both accuracy of user’s intention and speed.

-       Figure 6 reflects quantitative analysis of experiments on only 12 participants. On this number of participants, quantitative analysis is usually not an option.

-       Also Figure 7 relies on 12 participants for deciding about usability of the system, which is not the case.

Round 2

Reviewer 1 Report

This reviewer is still not satisfied with the responses to questions 1.2, 1.3, and 1.5. It is still unclear about the technique details being used and how they will address question 1.5 in practice. 

Reviewer 2 Report

The current version has been improved and the added sections helps readers to understand the paper better than the previous form. 

Round 3

Reviewer 1 Report

The authors have addressed my concerns. I have no further comments.